# SIMULATED+UNSUPERVISED LEARNING WITH ADAPTIVE DATA GENERATION AND BIDIRECTIONAL MAPPINGS

**Kangwook Lee**\*, **Hoon Kim**\* & **Changho Suh**
School of Electrical Engineering
KAIST
Daejeon, South Korea
{kw1jjang,gnsrla12,chsuh}@kaist.ac.kr

## ABSTRACT

Collecting a large dataset with high quality annotations is expensive and time-consuming. Recently, Shrivastava et al. (2017) propose *Simulated+Unsupervised (S+U) learning*: It first learns a mapping from synthetic data to real data, translates a large amount of labeled synthetic data to the ones that resemble real data, and then trains a learning model on the translated data. Bousmalis et al. (2017b) propose a similar framework that jointly trains a translation mapping and a learning model. While these algorithms are shown to achieve the state-of-the-art performances on various tasks, it may have a room for improvement, as they do not fully leverage flexibility of data simulation process and consider only the forward (synthetic to real) mapping. Inspired by this limitation, we propose a new S+U learning algorithm, which fully leverage the flexibility of data simulators and bidirectional mappings between synthetic and real data. We show that our approach achieves the improved performance on the gaze estimation task, outperforming (Shrivastava et al., 2017).

## 1 INTRODUCTION

Collecting a large annotated dataset is usually a very expensive and time-consuming task, and sometimes it is even infeasible. Recently, researchers have proposed the use of synthetic datasets provided by simulators to address this challenge. Not only synthetic datasets can be easily annotated but one can also generate an arbitrarily large amount of synthetic data. In addition to that, recent advances in computer technologies enabled synthesis of high-quality data.

Specifically, a variety of computer vision applications have benefitted from advanced computer graphics technologies. For instance, Wood et al. (2016) manipulate a 3D game graphics engine, called *Unity*, to synthesize photo-realistic images of human eye regions. Then, using a million of synthetic images with labels, they achieve the state-of-the-art performance on the cross-domain appearance-based gaze estimation task (Sugano et al., 2014). Another important application that is heavily benefitting from synthetic data is autonomous driving. A few recent works show that an infinite amount of realistic driving data can be collected from high-quality video games such as GTA V (Grand Theft Auto V) (Richter et al., 2016; Johnson-Roberson et al., 2017; Lee et al., 2017). Specifically, Richter et al. (2016) show that a semantic segmentation model trained only with synthetic data can even outperform the model trained with real data if the amount of synthetic data is large enough. In (Lee et al., 2017), the authors collect a synthetic dataset of vehicle accidents, which is hardly collectable from the real world, train an accident prediction model with the synthetic data, and then apply the trained model to a real accident dataset. As a result, they show that the model trained with large synthetic dataset can outperform the model trained within small real dataset. Further, researchers also propose the use of simulated environments for building algorithms for autonomous drones (Microsoft, 2017), autonomous truck driving (Im, 2017), etc.

---

\*These two authors contributed equally

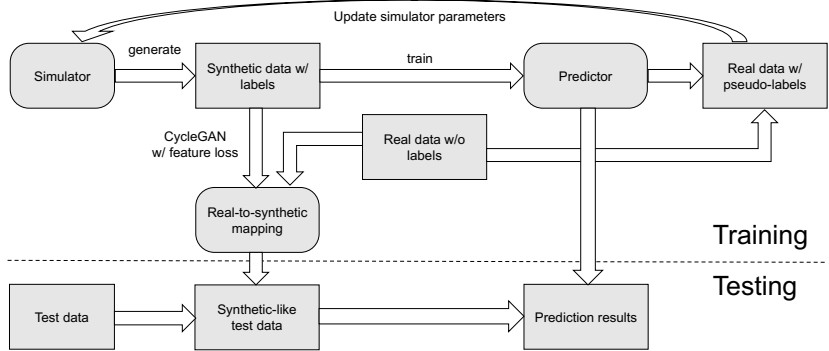

Figure 1: A block diagram of our S+U learning algorithm. The upper part of the diagram illustrates our training algorithm. First, we generate labeled synthetic data using a simulator, and then train a predictor on this data. Once the predictor is trained, we predict the labels of the unlabeled real data, which is available in the training phase. We call these predicted labels 'pseudo-labels'. Once the pseudo-labels are obtained, we compare the distribution of synthetic labels with that of pseudo-labels, and accordingly update the simulator parameters to reduce the gap. This procedure is repeated a few times until one achieves a small enough gap between the distributions. Once the synthetic dataset is finalized, we obtain a real-to-synthetic mapping between synthetic and real data. For testing, we first map test data to the synthetic domain, and then apply the predictor on it.

While the use of synthetic datasets is increasingly popular, it is not clear how one should systematically address a machine learning problem when simulated data is given with unlabeled real data. Recently, Shrivastava et al. (2017) propose *Simulated+Unsupervised (S+U) learning*, which is one the first methodologies that guides how one can use synthetic data to improve the performance of learning algorithms. They first learn a translation mapping from synthetic data to real data using a modified GAN (Generative Adversarial Networks) architecture (Goodfellow et al., 2014), map the synthetic data to the real-data domain, and then train a learning model with this mapped data. Using this methodology, they achieve the state-of-the-art performances for the cross-domain gaze estimation on the MPIIGaze dataset (Sugano et al., 2014). A contemporary paper by Bousmalis et al. (2017b) proposes a similar approach. They also show how one can accommodate both the task-specific loss and the domain-specific loss to further improve the quality of image transfer.

Even though the works of (Shrivastava et al., 2017; Bousmalis et al., 2017b) present interesting solutions to deal with simulated data, their solutions have some room for improvements for the following reasons. First, their approaches assume a fixed synthetic data, and does not leverage the flexibility of data simulation process. Since the data simulator can be freely manipulated, one may hope for further performance improvements. In addition to that, their approaches make a use of the forward (synthetic to real) mapping only, while recent works have shown the efficacy of bidirectional mappings between two domains (Zhu et al., 2017a; Kim et al., 2017). Inspired by these limitations, we propose a new S+U learning framework, which fully leverages both the flexibility of data simulators and bidirectional mappings between synthetic data and real data.

In this work, we propose a new S+U learning framework, consisting of three stages, visualized in Fig. 1. The first stage is where we fully leverage the flexibility of data simulators. We first predict the labels of the unlabeled real data, which is available in the training phase. We then update simulation parameters so that the synthetic label distribution matches the distribution of the predicted labels, and this procedure is repeated a few times. The second stage learns a bidirectional mapping between the synthetic data and the real data, using the cyclic image-to-image translation frameworks (Zhu et al., 2017a; Kim et al., 2017). To better preserve labels while translating images, we make a few modifications to the existing framework. In the last stage, we translate test (real) data to the synthetic domain, and then apply to them an inference model trained on the synthetic images. Note that our approach does not require additional training of the inference model. Our approach is superior if the backward mapping from real data to synthetic data is highly accurate and the inference model is well trained on the synthetic data. We show that our approach achieves the improved performance

on the gaze estimation task, outperforming the state of the art of (Shrivastava et al., 2017). Our contributions can be summarized as follows.

**Contributions**

1. A new end-to-end S+U learning algorithm is proposed (Sec. 2).
2. We develop a simple method of exploiting the flexibility of simulators (Sec. 2.2).
3. We propose a novel prediction method that first translates real images to synthetic images and then applies a prediction algorithm trained on synthetic images (Sec. 2.4).
4. Our method outperforms the state of the art on the gaze estimation task (Sec. 3).

## 2 OUR APPROACH

### 2.1 NOTATIONS

For notational simplicity, we denote the set of simulated images by a random variable $X \in \mathcal{X}$ and the real images by another random variable $Y \in \mathcal{Y}$. For the bidirectional mappings, we denote by $G_{\mathcal{X} \to \mathcal{Y}}$ the forward generator that maps simulated images to the real domain and by $G_{\mathcal{Y} \to \mathcal{X}}$ the backward generator that maps real images to the simulation domain. The content of an image, which we call *label of the image*, is denoted by $Z$. For instance, $Z$ denotes the digit contained in an image for the digit classification problem, and it denotes the eye gaze vector of an eye image for the eye gaze estimation problem. The label (or feature) extractor $F(\cdot)$ is a function that extracts the label (or feature) of a given image. Given an unlabeled real image of $Y$, we call $W = F(Y)$ the pseudo-label of the image.

### 2.2 ADAPTIVE DATA GENERATION BASED ON PSEUDO-LABELS

Let us first consider how one would build a simulator that can generate an arbitrary amount of synthetic data. For ease of explanation, imagine a simulator that generates synthetic images for the digit classification problem. A typical process of building a simulator is as follows. First, one chooses $P_Z$, the (synthetic) label distribution, arbitrarily or possibly aided with the prior knowledge on the target dataset. Then, the simulator specifies $P_{X|Z}$, the image distribution conditioned on labels. For instance, given $Z = 0$, a simulator might first draw a perfect ellipsoid and then add some pixel noise to reflect the diversity of images.

As illustrated above, a simulator is usually fully specified with the label distribution and the data (image) distribution conditioned on each label. A naïve choice of taking arbitrary distributions may result in a distributional gap between the synthetic dataset and real dataset, i.e., sample selection bias (Huang et al., 2007). Fortunately, for most existing simulators, it is easy to adjust the label distribution or $P_Z$. For instance, *UnityEyes* is a high-resolution 3D simulator that can render realistic human eye regions (Wood et al., 2016). The label is a $4$-dimensional vector, and the marginal distribution of each element can be specifeid by the user. See Sec. 3 for more details. On the other hand, the way an image is rendered given a label or $P_{X|Z}$ is not modifiable.

Motivated by this observation, we propose a simple iterative algorithm that can be used to find a good set of distribution parameters for the data generator. We note that the approach of (Shrivastava et al., 2017) does not adjust the data generation process, and stick with an arbitrary label distribution, say $P_{Z^{(0)}}$. Our algorithm is based on the novel use of *pseudo-labels* (Lee, 2013). Pseudo-label is originally proposed to tackle semi-supervised learning problems in which a large number of unlabeled data is provided with a small number of labeled data: Once a coarse predictor is trained with the small labeled dataset, one can assign pseudo-labels to the large unlabeled dataset using the prediction results of the coarse predictor.

We now describe our algorithm, visualized in Fig. 2. It first starts with an arbitrary label distribution $P_{Z^{(0)}}$ (and hence an arbitrary data distribution $P_{X^{(0)}}$), and then trains a predictor $F^{(0)}$ using the generated data. Once the predictor $F^{(0)}$ is trained, it computes the empirical distribution of the pseudo-labels of the unlabeled real dataset, i.e., $W^{(0)} = F^{(0)}(Y)$. Finally, it finds the new label distribution $P_{Z^{(1)}}$ for the simulator by minimizing the distance to the pseudo-label distribution $P_{W^{(0)}}$

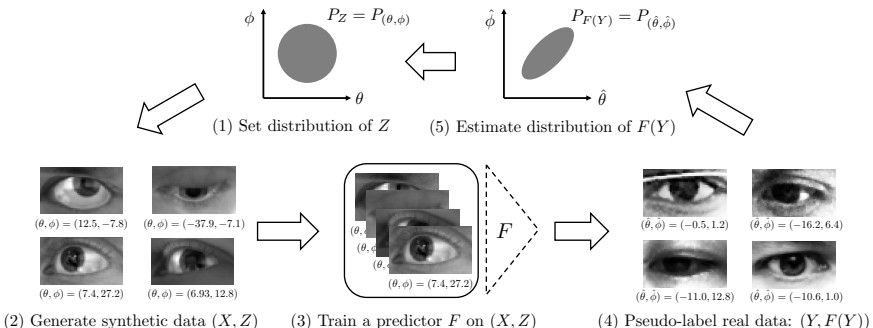

(1) Set distribution of $Z$     (5) Estimate distribution of $F(Y)$

(2) Generate synthetic data $(X, Z)$     (3) Train a predictor $F$ on $(X, Z)$     (4) Pseudo-label real data: $(Y, F(Y))$

Figure 2: An overview of our adaptive data generation process. For the ease of explanation, consider the eye gaze estimation problem where the goal is to estimate a 2-dimensional gaze vector $Z = (\theta, \phi)$ given an image of the eye region. In the beginning, the simulator arbitrarily initializes the label distribution, say $P_Z$. It then draws random labels according to the label distribution, and generates corresponding images $X$ according to some rendering rules. Using this synthetic dataset, it trains a predictor $F$, and then predicts the gaze vectors of each real image (initially unlabeled), i.e., annotates each image $Y$ with a *pseudo-label* $W = (F_\theta(Y), F_\phi(Y))$. The last stage estimates the pseudo-label distribution $P_W$, which is used as the initial distribution of the subsequent iteration.

in some metric, and then accordingly updates the simulator parameters.[1] This procedure can be repeated a few times.

## 2.3 LABEL-PRESERVING IMAGE-TO-IMAGE TRANSLATION

CycleGAN is an unsupervised image-to-image translation method based on GANs (Zhu et al., 2017a). In this section, we first observe that image content, i.e., *labels of the image*, may alter when translated via the original CycleGAN framework. In order to mitigate this problem, we propose a slight modification to the CycleGAN framework by employing the concept of "content representation" (Gatys et al., 2016). The key idea is simple: Given a label (content) extractor $F(\cdot)$, we simply regularize the difference between the labels of the original image and its translated image. Note that this idea is also called perceptual loss or feature matching (Gatys et al., 2017; Bousmalis et al., 2017a), and a similar idea has been applied in other works on unsupervised image-to-image translations (Shrivastava et al., 2017; Taigman et al., 2016).

To illustrate the limitation of the CycleGAN framework, we first formally describe the CycleGAN framework. The goal is to find $G_{\mathcal{X} \to \mathcal{Y}}$, the forward generator from $\mathcal{X}$ to $\mathcal{Y}$, and $G_{\mathcal{Y} \to \mathcal{X}}$, the backward generator from $\mathcal{Y}$ to $\mathcal{X}$. These generators, or translation mappings, take images from the source domain and convert them to the target domain without altering the image content. Using the GAN framework of Goodfellow et al. (2014), we train these generators jointly with discriminators: $D_{\mathcal{Y}}$ ($D_{\mathcal{X}}$) outputs the probability that the given image is from $\mathcal{Y}$ ($\mathcal{X}$). Using these notations, the standard GAN loss for the forward mapping can be written as

$$\mathcal{L}_{\text{GAN, forward}} = \mathbb{E}_Y[\log D_{\mathcal{Y}}(Y)] + \mathbb{E}_X[\log(1 - D_{\mathcal{Y}}(G_{\mathcal{X} \to \mathcal{Y}}(X)))]. \tag{1}$$

The backward GAN loss is similarly defined with $(X, \mathcal{X})$ and $(Y, \mathcal{Y})$ being interchanged. Further, the CycleGAN framework has an additional loss term, called the cycle-consistency loss:

$$\mathcal{L}_{\text{cyc}} = \mathbb{E}_X[\|G_{\mathcal{Y} \to \mathcal{X}}(G_{\mathcal{X} \to \mathcal{Y}}(X)) - X\|] + \mathbb{E}_Y[\|G_{\mathcal{X} \to \mathcal{Y}}(G_{\mathcal{Y} \to \mathcal{X}}(Y)) - Y\|], \tag{2}$$

where $\|\cdot\|$ can be an arbitrary norm, e.g., $\ell_1$ norm. This loss essentially imposes a restriction on the forward/backward mappings so that $G_{\mathcal{Y} \to \mathcal{X}}(G_{\mathcal{X} \to \mathcal{Y}}(X)) \simeq X$, and similarly, $G_{\mathcal{X} \to \mathcal{Y}}(G_{\mathcal{Y} \to \mathcal{X}}(Y)) \simeq Y$. Combining these terms and omitting the function arguments for simplicity, the overall loss function is as follows:

$$\mathcal{L} = \mathcal{L}_{\text{GAN, forward}} + \mathcal{L}_{\text{GAN, backward}} + \lambda_{\text{cyc}} \mathcal{L}_{\text{cyc}}, \tag{3}$$

---

[1]In our experiments with the UnityEye simulator, we simply employ the moment matching algorithm since the simulator only allows us to specify the first and second moments of the label distribution. However, in general, one may choose an arbitrary optimization algorithm for updating the label distribution.

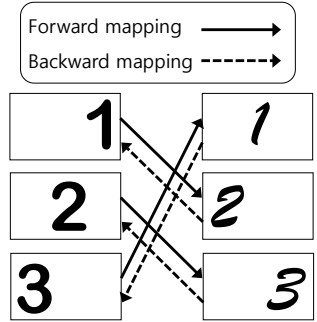

Figure 3: A toy example where perfect cycle-consistency is achieved but labels are permuted.

where $\lambda_{\text{cyc}}$ is the regularization parameter. Then, one can apply the standard training methodology of GANs to find $G$'s that minimize the loss function and $D$'s that maximize it.

While the original CycleGAN is observed to well preserve labels under certain scenarios, its theoretical understanding is still missing. Further, the authors of the CycleGAN paper also acknowledged the potential limitation of the original approach: they made a remark that their method sometimes permutes the labels for tree and building when applied to the cityscapes photos, and provided failure cases (Zhu et al., 2017b). Indeed, we also observe that it frequently fails to maintain labels: See Sec. B for some examples.

Indeed, it can be explained via a simple example why the cycle-consistency loss alone can fail to preserve labels. See Fig. 3 for a toy example where the goal is to learn an image-to-image translation rule between digit images in Arial font ($\mathcal{X}$) and those in Script font ($\mathcal{Y}$), where the label is the number present in the image. Consider the $G_{\mathcal{X} \to \mathcal{Y}}$ and $G_{\mathcal{Y} \to \mathcal{X}}$ shown as arrows in the figure. Note that the labels, or the digits contained in images, are not preserved but permuted. However, observe that these mappings result in perfect cycle-consistency, i.e., $\mathcal{L}_{\text{cyc}} = 0$. Further, if the discriminators $D_{\mathcal{X}}$ and $D_{\mathcal{Y}}$ are perfect, $\mathcal{L}_{\text{GAN, forward}} = \mathcal{L}_{\text{GAN, backward}} = 0$. This implies that the zero loss can be achieved even when labels are not preserved at all. In general, with $n$ samples in each domain, there are at least $n! - 1$ pairs of mappings that attain zero loss without preserving labels.

As mentioned in the beginning of the section, we employ *the feature-consistency loss* in order to better preserve labels. Specifically, we use the bidirectional feature-consistency loss, defined as

$$\mathcal{L}_{\text{feature}} = \mathbb{E}_X[\|F(G_{\mathcal{X} \to \mathcal{Y}}(X)) - F(X)\|] + \mathbb{E}_Y[\|F(G_{\mathcal{Y} \to \mathcal{X}}(Y)) - F(Y)\|], \quad (4)$$

where $\|\cdot\|$ is an appropriate norm of the feature space. We then use the following total loss function:

$$\mathcal{L} = \mathcal{L}_{\text{GAN, forward}} + \mathcal{L}_{\text{GAN, backward}} + \lambda_{\text{cyc}} \mathcal{L}_{\text{cyc}} + \lambda_{\text{feature}} \mathcal{L}_{\text{feature}}. \quad (5)$$

Going back to the toy example, assume that we are given with a perfect digit classifier $F$. With the feature consistency loss term added to the objective function, the mappings shown in Fig. 3 will incur non-zero loss. In order to achieve zero loss, one must find the correct bidirectional mapping between the two domain that preserve labels. Hence, generators will attempt at maintaining the labels between image pairs to minimize the loss.

### 2.4 PREDICTION WITH BACKWARD TRANSLATION

After the translation mappings are obtained, there are two possible approaches that one can take. The first approach, which is the first simulated+unsupervised learning approach to image-to-image translations, is proposed by Shrivastava et al. (2017); Bousmalis et al. (2017b). It first translates all the labeled synthetic data using the forward mapping $G_{\mathcal{X} \to \mathcal{Y}}$, and then train a model on this real-like (labeled) dataset. Both works show that this approach significantly outperforms the previous state of the arts for diverse tasks. See Fig. 4a for visual illustration.

The second approach, which we propose in this work, relies on the backward translation, i.e., $G_{\mathcal{Y} \to \mathcal{X}}$. As described in the previous section, leveraging simulators, one can generate an arbitrarily large amount of synthetic data, and then train an efficient model on it. Hence, assuming perfect

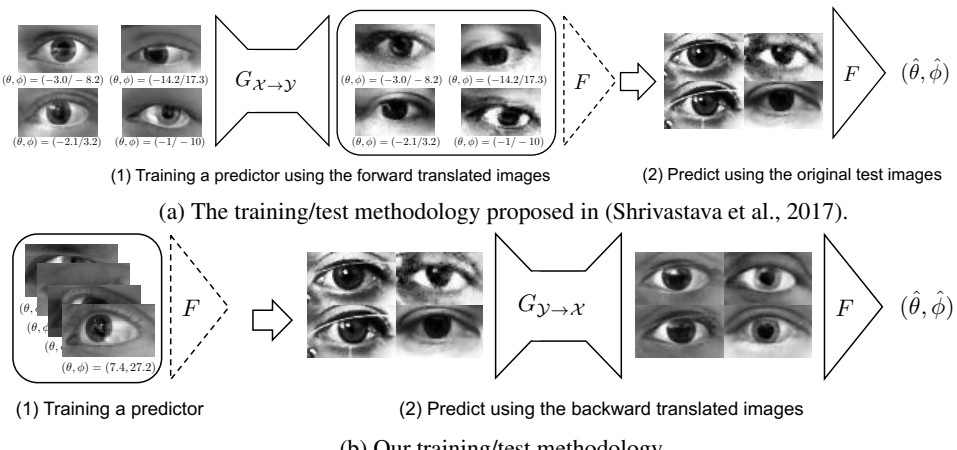

(1) Training a predictor using the forward translated images      (2) Predict using the original test images

(a) The training/test methodology proposed in (Shrivastava et al., 2017).

(1) Training a predictor      (2) Predict using the backward translated images

(b) Our training/test methodology.

Figure 4: Comparison of our methodology with the original S+U learning approach (Shrivastava et al., 2017; Bousmalis et al., 2017b).

backward translation, one can first translate real data to synthetic data, and then apply the model (trained on synthetic data) to the mapped data. See Fig. 4b for visual illustration.

Indeed, in most practically relevant cases, the amount of available synthetic data is much larger than that of real data, and hence one can easily train a highly accurate predictor at least in the synthetic domain (but not necessarily in the real domain). Another advantage of this approach is that one does not have to train a new model when a new real dataset arrives, while the first approach needs to retrain the prediction model for each real dataset. We remark that this approach requires an additional computational cost for backward translation during prediction. However, the additional computational cost is usually not prohibitive compared to the complexity of the prediction model.

## 3 EXPERIMENTS

### 3.1 CROSS-DOMAIN APPEARANCE-BASED GAZE ESTIMATION

In this section, we apply our methodology to tackle the cross-domain appearance-based gaze estimation problem, and evaluate its performance on the the MPIIGaze dataset (Zhang et al., 2015). The goal of this problem is to estimate the gaze vector given an image of a human eye region using the data collected from a different domain.

To generate a synthetic dataset of labeled human eye images, we employ *UnityEyes*, a high-resolution 3D simulator that renders realistic human eye regions (Wood et al., 2016). For each image, UnityEyes draws the pitch and yaw of the eyeball and camera, uniformly at random, and then renders the corresponding eye region image. More specifically, the random distributions are specified by 8 input parameters $(\theta_p, \theta_y, \phi_p, \phi_y, \delta\theta_p, \delta\theta_y, \delta\phi_p, \delta\phi_y)$: The first two are the expected values of eyeball pitch and yaw, and the following two are the expected values of camera pitch and yaw; And the other four are the half-widths of the distributions. As a default setting, UnityEyes uses the following distribution parameters: $(\theta_p, \theta_y, \phi_p, \phi_y, \delta\theta_p, \delta\theta_y, \delta\phi_p, \delta\phi_y) = (0°, 0°, 0°, 0°, 30°, 30°, 20°, 40°)$. More details on how UnityEyes render eye regions can be found in (Wood et al., 2016).

We first evaluate the performance of our data generation algorithm based on pseudo-labels. Using the 380k images generated with UnityEyes with the default parameters, we train a simple gaze estimation network, with which we annotate the real dataset with pseudo-labels. We then obtain the horizontal and vertical statistics of the gaze vectors, and accordingly adjust the UnityEyes parameters to make the means and variances coincide. Since there are many free parameters, we reduce the number of free parameters by considering the following specific classes of distributions: $\theta_p = \phi_p, \theta_y = \phi_y, \delta\theta_p = \delta\phi_p, \delta\theta_y = \delta\phi_y$.[2] We run the data adaption algorithm for 4 iterations,

---

[2]We note that the choice of parameter reduction does not much affect the overall performance: We test a different reduction method and report the corresponding results in the appendix.

and report in Table 1 the sequences of means and standard deviations, relative to the ground truth. Further, we also run our algorithm starting from each label distribution and report the validation/test errors in the table. We observe that after the second iteration of the algorithm, the validation/test errors are minimized. Note that in this experiment, the validation error increases after the second iteration in our experiment, implying that the first stage of our algorithm may not converge well in practice. Hence, one needs to use a small validation set to choose which iteration to proceed with.

In the rest of this section, we report the results based on the output of the second iteration. Using the newly generated synthetic dataset together with the unlabeled real dataset, we then train a cyclic image-to-image translation mapping with the new loss function (5). For the choice of the feature, we set the identity mapping as the feature extractor since two domains share the high-level structure and differ only in detailed expressions. We use the $\ell_1$ norm for the feature-consistency term. For the choice of regularization parameters, we test the performance of our algorithm with $\lambda_{\text{cyc}} \in \{0, 1, 5, 10, 50\}$ and $\lambda_{\text{feature}} \in \{0, 0.1, 0.5, 1.0\}$. The test results are summarized in Table 3 in Sec. C.1. As a result, we observe that $\lambda_{\text{cyc}} = 10$ and $\lambda_{\text{feature}} = 0.5$ achieved the best performance.

Shown in Fig. 5 are some examples of the translation results. Here, we compare the CycleGAN trained only with the cycle-consistency loss and that trained with both the cycle-consistency loss and the feature-consistency loss. The first column shows the input real images, the next two columns are the translated images using the output of the former approach, and the last two columns are the outputs of the latter approach. It is clear that the the learned mappings satisfy near-perfect cycle-consistency by comparing $Y$'s with $G_{\mathcal{X} \to \mathcal{Y}}(G_{\mathcal{Y} \to \mathcal{X}}(Y))$'s. However, the gaze vector is not preserved when real images are translated to synthetic images. On the other hand, when both consistencies are enforced, the gaze vector does not alter, maintaining the label information, showing the necessity of the feature-consistency loss term.

Summarized in Table 2 are the experimental results along with the state of the arts reported in the literature. We first run the first two stages of our algorithm using the optimal regularization parameters. To see the gain of the adaptive data generation and the bidirectional mapping, we first apply the forward mapping to the simulated data to obtain refined synthetic (RS) images. Then, we train a new predictor on these RS images, and then apply this predictor to the test images. Note that this can be viewed as a combination of the first two stages of our algorithm and the original S+U learning approach. Using this approach, we obtain the test error of 7.71, outperforming the state-of-the-art performance of Shrivastava et al. (2017). Note that this performance improvement is due to our adaptive data generation algorithm and the use of bidirectional mapping. Further, our algorithm is trained only with 380k images while the results of Shrivastava et al. (2017) is trained with 1200k synthetic images, demonstrating the sample-efficiency of our algorithm. Further, instead of refining synthetic images and training a new predictor on that, we apply the backward mapping to the real data and apply the predictor trained on the synthetic data. As a result, we achieve the test error of 7.60, showing that our backward mapping can further improve the prediction performance.

## 3.2 IMPLEMENTATION DETAILS

For our experiments, we slightly modify the CycleGAN framework of Zhu et al. (2017a). The generator network, $G$, is modified to take input image of $36 \times 60$, and the rest of the $G$ is identical to that of the original CycleGAN architecture with 6 blocks. The discriminator network, $D$ is identical

Table 1: Experimental results of the adaptive data generation algorithm. Here, $\ell$ denotes the number of iterations. The first four rows are sequences of means and standard deviations of $(\Theta^{(\ell)}, \Phi^{(\ell)})$, compared with the true label distribution $(\Theta^\star, \Phi^\star)$. The last row is for the validation/test errors.

| | $\ell = 0$ | $\ell = 1$ | $\ell = 2$ | $\ell = 3$ | $\ell = 4$ |
|---|---|---|---|---|---|
| $\|\mathbb{E}[\Theta^{(\ell)}] - \mathbb{E}[\hat{\Theta}^\star]\|$ | 8.54 | 3.00 | 1.04 | 0.96 | 0.36 |
| $\|\sigma(\Theta^{(\ell)}) - \sigma(\hat{\Theta}^\star)\|$ | 14.87 | 4.18 | 0.47 | 1.30 | 4.63 |
| $\|\mathbb{E}[\Phi^{(\ell)}] - \mathbb{E}[\hat{\Phi}^\star]\|$ | 0.31 | 0.02 | 0.00 | 0.01 | 0.01 |
| $\|\sigma(\Phi^{(\ell)}) - \sigma(\hat{\Phi}^\star)\|$ | 20.00 | 2.30 | 1.65 | 3.58 | 7.64 |
| Val./Test error | 22.88/22.6 | 9.09/8.80 | **7.42/7.60** | 8.07/8.04 | 9.10/9.21 |

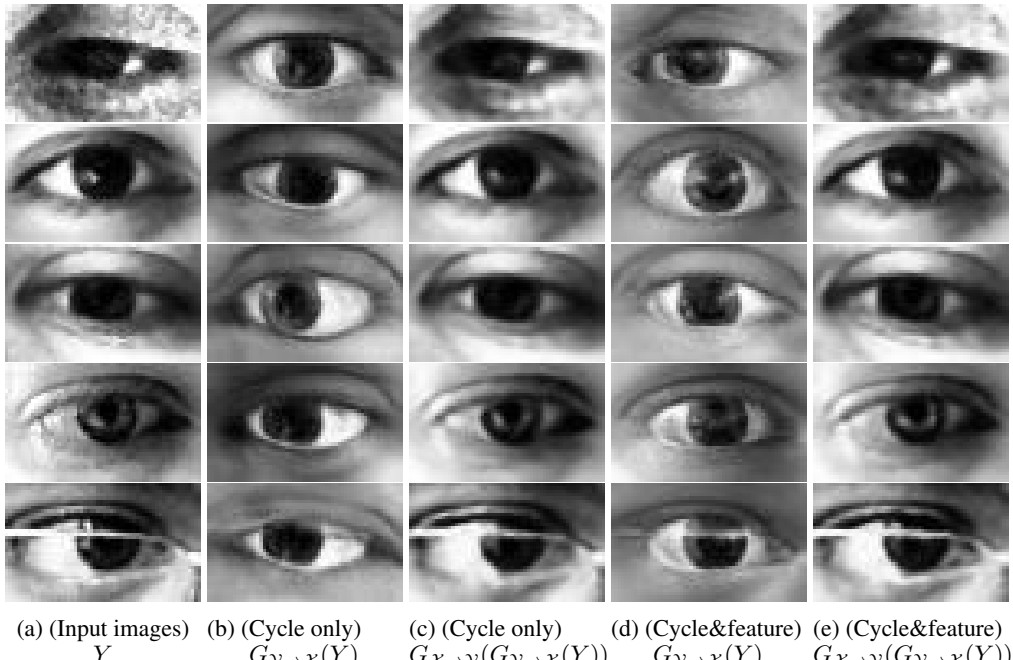

(a) (Input images)   (b) (Cycle only)   (c) (Cycle only)   (d) (Cycle&feature)   (e) (Cycle&feature)
$Y$   $G_{\mathcal{Y}\to\mathcal{X}}(Y)$   $G_{\mathcal{X}\to\mathcal{Y}}(G_{\mathcal{Y}\to\mathcal{X}}(Y))$   $G_{\mathcal{Y}\to\mathcal{X}}(Y)$   $G_{\mathcal{X}\to\mathcal{Y}}(G_{\mathcal{Y}\to\mathcal{X}}(Y))$

Figure 5: Some examples of the translation results (real $\to$ synthetic $\to$ real). The first column is the original real image, say $Y$. The second and third columns are $G_{\mathcal{Y}\to\mathcal{X}}(Y)$ and $G_{\mathcal{X}\to\mathcal{Y}}(G_{\mathcal{Y}\to\mathcal{X}}(Y))$, where $G$'s are trained with the cyclic consistency term. The last two columns are $G_{\mathcal{Y}\to\mathcal{X}}(Y)$ and $G_{\mathcal{X}\to\mathcal{Y}}(G_{\mathcal{Y}\to\mathcal{X}}(Y))$, where $G$'s are trained with both the cyclic consistency term and the feature consistency term. Both approaches achieve nearly perfect cycle-consistency; However, only the latter approach maintains the structure of the images when translated to the other domain.

Table 2: Comparison of the state of the arts on the MPIIGaze dataset. The error is the mean angle error in degrees. The parameter $\alpha$ denotes the learning rate used for training the predictor. In the second and third columns, 'R' denotes 'Real', 'S' denotes 'Synthetic', 'RS' denotes 'Refined Synthetic', and 'RR' denotes 'Refined Real'. Our approaches with hyperparameters ($\ell = 2, \lambda_{\mathrm{cyc}} = 10, \lambda_{\mathrm{feature}} = 0.5$) achieve the state-of-the-art performances.

| Method | Trained on | Tested with | Error |
|---|---|---|---|
| Support Vector Regression (SVR) (Schneider et al., 2014) | R | R | 16.5 |
| Adaptive Linear Regression (ALR) (Lu et al., 2014) | R | R | 16.4 |
| kNN w/ UT Multiview (Zhang et al., 2015) | R | R | 16.2 |
| Random Forest (RF) (Sugano et al., 2014) | R | R | 15.4 |
| CNN w/ UT Multiview (Zhang et al., 2015) | R | R | 13.9 |
| CNN w/ UnityEyes (Shrivastava et al., 2017) | S | R | 11.2 |
| kNN w/ UnityEyes (Wood et al., 2016) | S | R | 9.9 |
| SimGAN (Shrivastava et al., 2017) | RS | R | 7.8 |
| Ours (Adaptive data generation + Bidirectional + Forward) | RS | R | **7.71** |
| Ours (Adaptive data generation + Bidirectional + Backward) | S | RR | **7.60** |

to that of the original CycleGAN framework. For detailed information, we refer the readers to (Zhu et al., 2017a). The CycleGAN was trained with batch size of $64$ and learning of $2 \times 10^{-4}$.

The eye gaze prediction network is designed based on the architecture proposed in (Shrivastava et al., 2017). The input is a $36 \times 60$ gray-scale image that is passed through $5$ convolutional layers followed by $3$ fully connected layers, the last one encoding the $3$-dimensional (normalized) gaze vector: (1) Conv3x3, $32$ feature maps, (2) Conv3x3, $32$ feature maps, (3) Conv3x3, $64$ feature maps, (4) MaxPool3x3, stride $= 2$, (5) Conv3x3, $80$ feature maps, (6) Conv3x3, $192$ feature maps, (7)

MaxPool2x2, stride $= 2$, (8) FC9600, (9) FC1000, (10) FC3, (11) $\ell_2$ normalization (12) $\ell_2$ loss. The predictor network is trained with batches of size $512$, until the validation error converges.

## 4 CONCLUSION

In this work, we propose a new S+U learning algorithm, which fully exploits the flexibility of simulators and the recent advances in learning bidirectional mappings, and show that it outperforms the state-of-the-art performance on the gaze estimation task.

We conclude the paper by enumerating a few open problems. In our experiments, we arbitrarily choose the feature for the consistency term. We, however, observe that the choice of feature mapping significantly affects the performance of the algorithm. Thus, one needs to study how the optimal feature mapping can be designed for different tasks.

In this work, we have separated learning of cross-domain translation mappings from learning of a prediction algorithm. One interesting open question is whether one can jointly train these two components and potentially outperform the current separation-based approach.

Another interesting topic to study is the regularization techniques for the adaptive data generation method. For instance, if the real data provided in the training set is not representative enough, our approach, which tries to match the synthetic label distribution with the real one, may not be able to generalize well on unseen data. One may address this limitation by incorporating prior knowledge about the label distributions or manually tune the simulation parameters. A thorough study is needed to understand how one could obtain diverse synthetic images via such methods in a systematic way.

Further, our adaptive data generation method assumes that the predictor trained on simulated data works quite well on real data. In other words, if the predictor trained solely on simulated data provide completely wrong pseudo-labels, matching the synthetic label distribution with the pseudo-label distribution may not be helpful at all. For instance, when we pseudo-label the images in the Street View House Numbers (SVHN) dataset using a digit classifier that is trained on the MNIST dataset, the resultant pseudo-label distribution is observed to be useless to refine the synthetic label distribution (Netzer et al., 2011; LeCun et al., 1998). It is an interesting open question whether or not one can devise a similar adaptive data generation method for such cases.

Building a differentiable data generator is also an interesting topic (Graves et al., 2014; Gaunt et al., 2017; Feser et al., 2016). It is well known that neural networks with external memory resources is a differentiable Turing Machine or differentiable Von Neumann architecture (Graves et al., 2014). Further, researchers have proposed the use of differentiable programming language with neural networks Gaunt et al. (2017); Feser et al. (2016). Indeed, the adaptive data generation algorithm proposed in this work can be viewed as an extremely limited way of adjusting the way synthetic data is generated. If the data generator can be written in a differentiable language, one could possibly jointly optimize the synthetic data generator together with the other components such as translation networks and prediction networks, potentially achieving an improved performance.

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

## A    ADDITIONAL EXPERIMENTAL RESULTS ON EYE GAZE ESTIMATION

In this section, we provide additional qualitative results on the eye gaze estimation task. See Fig. 6.

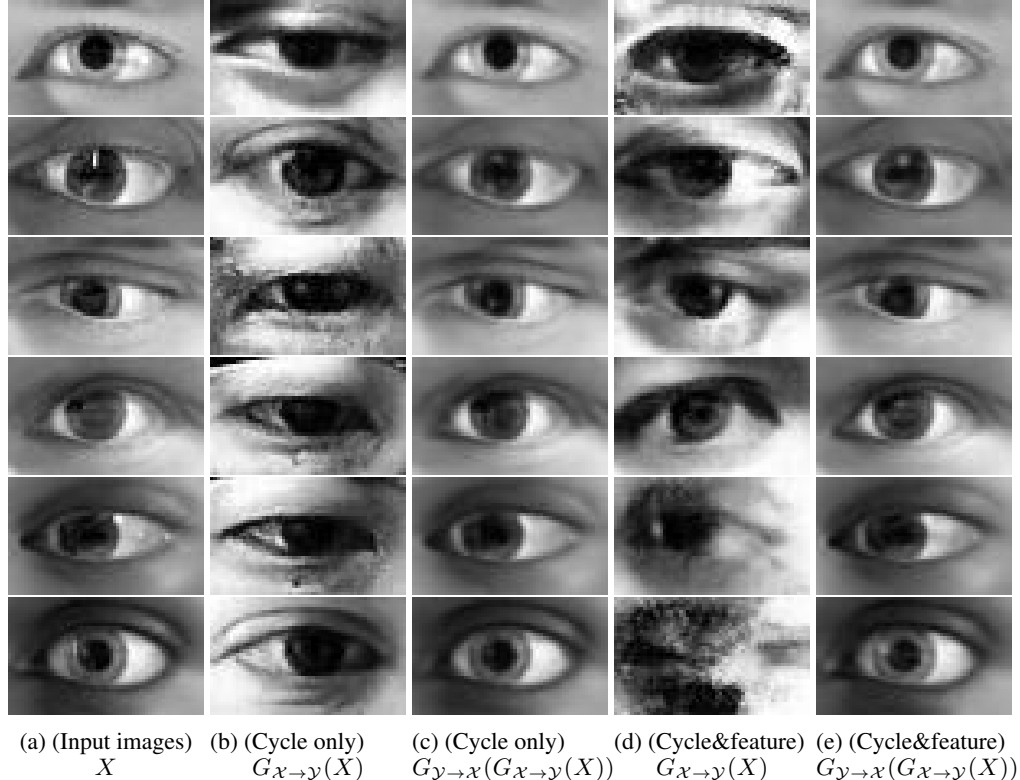

(a) (Input images)  (b) (Cycle only)  (c) (Cycle only)  (d) (Cycle&feature)  (e) (Cycle&feature)
$X$  $G_{\mathcal{X}\to\mathcal{Y}}(X)$  $G_{\mathcal{Y}\to\mathcal{X}}(G_{\mathcal{X}\to\mathcal{Y}}(X))$  $G_{\mathcal{X}\to\mathcal{Y}}(X)$  $G_{\mathcal{Y}\to\mathcal{X}}(G_{\mathcal{X}\to\mathcal{Y}}(X))$

Figure 6: Some examples of the translation results (synthetic $\to$ real $\to$ synthetic). The first column is the original synthetic image, say $X$. The second and third columns are $G_{\mathcal{X}\to\mathcal{Y}}(X)$ and $G_{\mathcal{Y}\to\mathcal{X}}(G_{\mathcal{X}\to\mathcal{Y}}(X))$, where $G$'s are trained with the cyclic consistency term. The last two columns are $G_{\mathcal{X}\to\mathcal{Y}}(X)$ and $G_{\mathcal{Y}\to\mathcal{X}}(G_{\mathcal{X}\to\mathcal{Y}}(X))$, where $G$'s are trained with both the cyclic consistency term and the feature consistency term.

# B ADDITIONAL EXPERIMENTAL RESULTS ON GTA DATA

We learn a bidirectional mapping between the driving images collected from the video game GTA V and the KITTI driving dataset (Geiger et al., 2012). Here, we use the last fully connected layer of `resnet18` as the feature mapping (He et al., 2016). Shown in Fig. 7 and Fig. 8 are some qualitative mapping results, proving the efficacy of the feature-consistency loss.

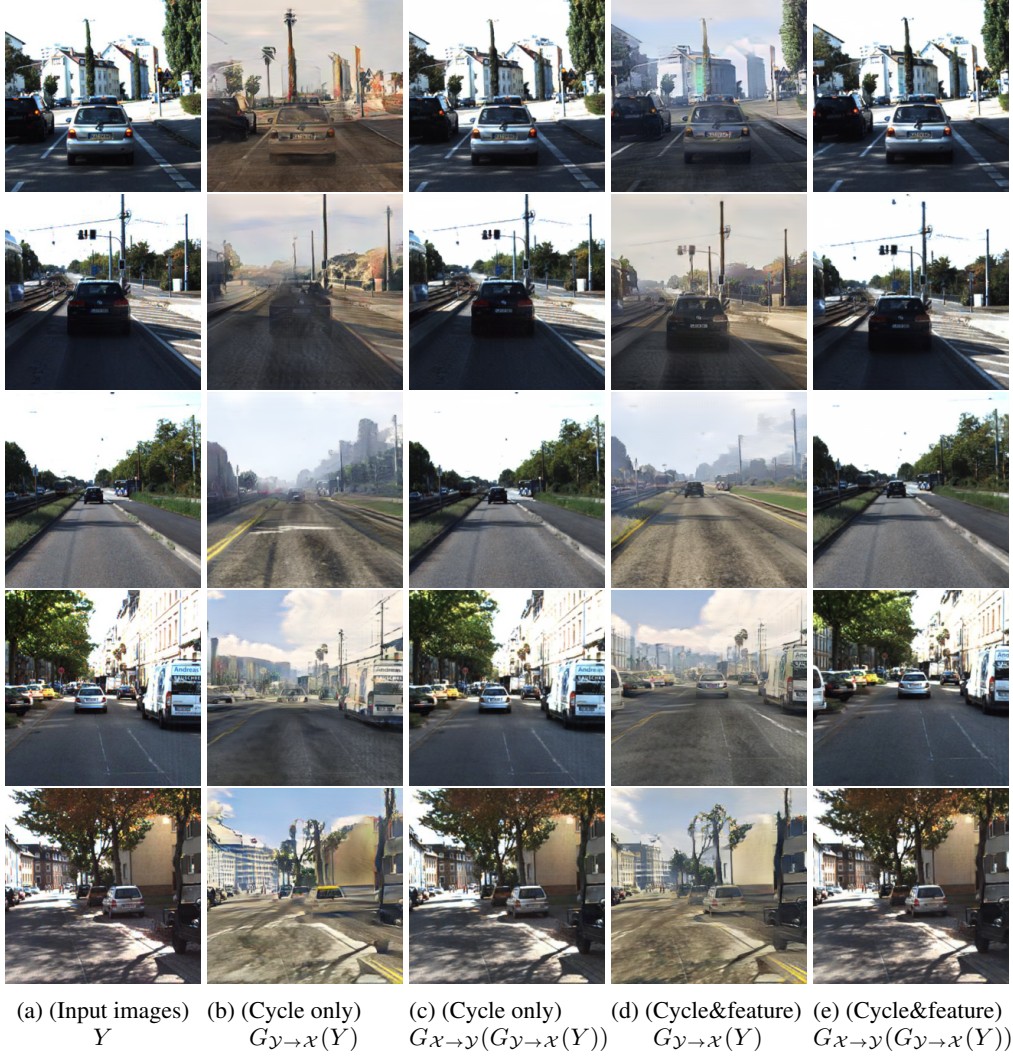

| (a) (Input images) | (b) (Cycle only) | (c) (Cycle only) | (d) (Cycle&feature) | (e) (Cycle&feature) |
|---|---|---|---|---|
| $Y$ | $G_{\mathcal{Y}\to\mathcal{X}}(Y)$ | $G_{\mathcal{X}\to\mathcal{Y}}(G_{\mathcal{Y}\to\mathcal{X}}(Y))$ | $G_{\mathcal{Y}\to\mathcal{X}}(Y)$ | $G_{\mathcal{X}\to\mathcal{Y}}(G_{\mathcal{Y}\to\mathcal{X}}(Y))$ |

Figure 7: Some examples of the translation results (real $\to$ synthetic $\to$ real). The first column is the original real image, say $Y$. The second and third columns are $G_{\mathcal{Y}\to\mathcal{X}}(Y)$ and $G_{\mathcal{X}\to\mathcal{Y}}(G_{\mathcal{Y}\to\mathcal{X}}(Y))$, where $G$'s are trained with the cyclic consistency term. The last two columns are $G_{\mathcal{Y}\to\mathcal{X}}(Y)$ and $G_{\mathcal{X}\to\mathcal{Y}}(G_{\mathcal{Y}\to\mathcal{X}}(Y))$, where $G$'s are trained with both the cyclic consistency term and the feature consistency term. Both approaches achieve nearly perfect cycle-consistency; However, only the latter approach maintains the structure of the images when translated to the other domain.

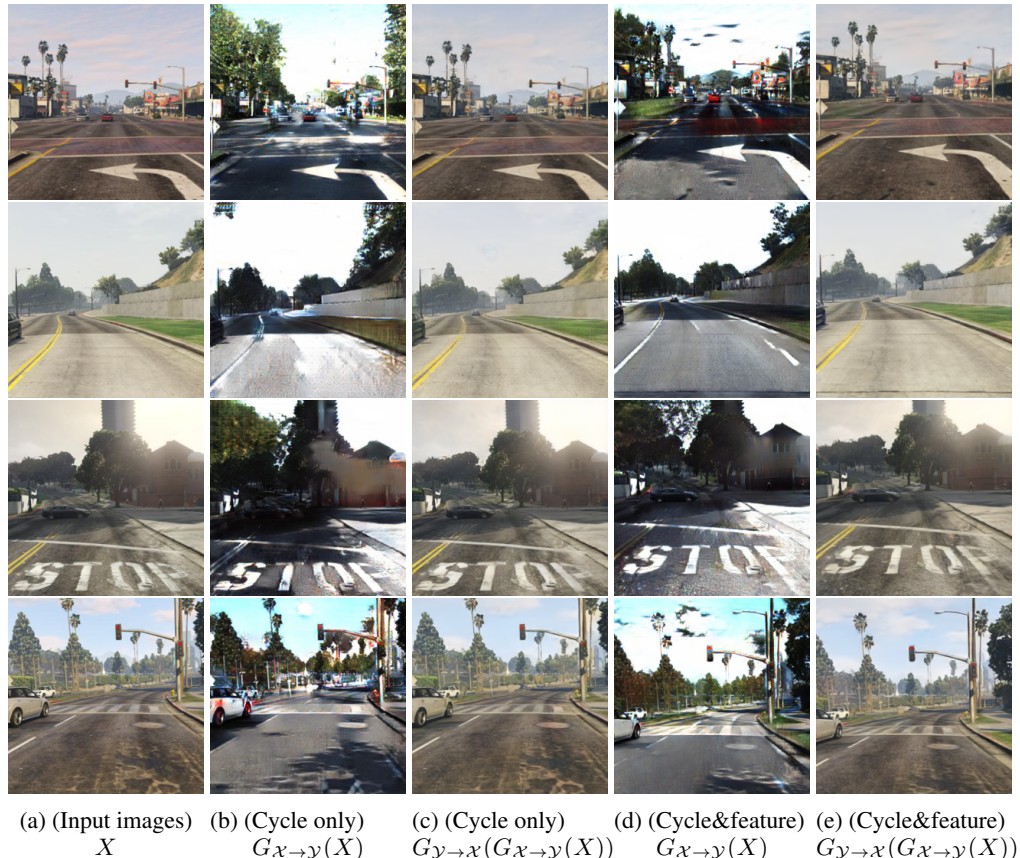

| (a) (Input images) | (b) (Cycle only) | (c) (Cycle only) | (d) (Cycle&feature) | (e) (Cycle&feature) |
| :---: | :---: | :---: | :---: | :---: |
| $X$ | $G_{\mathcal{X} \to \mathcal{Y}}(X)$ | $G_{\mathcal{Y} \to \mathcal{X}}(G_{\mathcal{X} \to \mathcal{Y}}(X))$ | $G_{\mathcal{X} \to \mathcal{Y}}(X)$ | $G_{\mathcal{Y} \to \mathcal{X}}(G_{\mathcal{X} \to \mathcal{Y}}(X))$ |

Figure 8: Some examples of the translation results (synthetic $\to$ real $\to$ synthetic). The first column is the original synthetic image, say $X$. The second and third columns are $G_{\mathcal{X} \to \mathcal{Y}}(X)$ and $G_{\mathcal{Y} \to \mathcal{X}}(G_{\mathcal{X} \to \mathcal{Y}}(X))$, where $G$'s are trained with the cyclic consistency term. The last two columns are $G_{\mathcal{X} \to \mathcal{Y}}(X)$ and $G_{\mathcal{Y} \to \mathcal{X}}(G_{\mathcal{X} \to \mathcal{Y}}(X))$, where $G$'s are trained with both the cyclic consistency term and the feature consistency term.

## C  ADDITIONAL EXPERIMENTS

### C.1  OPTIMAL CHOICE OF REGULARIZATION PARAMETERS

When training the bidirectional mappings, one needs to choose the values of $\lambda_{\text{cyc}}$ and $\lambda_{\text{feature}}$. In this section, we report the test errors measured with different choices of these regularization parameters in Table 3. As a result, we observe that the choice $(\lambda_{\text{cyc}} = 10, \lambda_{\text{feature}} = 0.5)$ obtains the best performance.

Table 3: Test results with different pairs of regularization parameters $\lambda_{\text{cyc}}$ and $\lambda_{\text{feature}}$.

| $\lambda_{\text{cyc}} \backslash \lambda_{\text{feature}}$ | 0 | 0.1 | 0.5 | 1.0 |
| :---: | :---: | :---: | :---: | :---: |
| 0 | 9.26 | 9.27 | 8.73 | 8.05 |
| 1 | 9.16 | 8.46 | 7.85 | 7.97 |
| 5 | 8.36 | 7.63 | 7.70 | 7.71 |
| 10 | 7.70 | 8.01 | **7.60** | 7.65 |
| 50 | 8.64 | 8.89 | 7.78 | 7.77 |

## C.2 PARAMETER REDUCTION

Since the number of observed variables is larger than the number of simulation parameters, we reduced the number of free parameters by using following reduction: $\theta_p = \phi_p, \theta_y = \phi_y, \delta\theta_p = \delta\phi_p, \delta\theta_y = \delta\phi_y$. Here, we conduct an additional experiment with a different reduction method: $2\theta_p = \phi_p, 2\theta_y = \phi_y, 2\delta\theta_p = \delta\phi_p, 2\delta\theta_y = \delta\phi_y$. We adapt the data distribution with this new reduction method for 4 times. As a result, we obtain the same test error performance with this different reduction method, implying that the reduction method does not much affect the overall performance.

Table 4: Experimental results of the adaptive data generation algorithm with a different reduction method. Here, $\ell$ denotes the number of iterations.

| | $\ell = 0$ | $\ell = 1$ | $\ell = 2$ | $\ell = 3$ | $\ell = 4$ |
|---|---|---|---|---|---|
| $\|\mathbb{E}[\Theta^{(\ell)}] - \mathbb{E}[\hat{\Theta}^\star]\|$ | 8.54 | 2.97 | 1.94 | 0.14 | 0.90 |
| $\|\sigma(\Theta^{(\ell)}) - \sigma(\hat{\Theta}^\star)\|$ | 14.87 | 4.25 | 0.43 | 2.68 | 4.63 |
| $\|\mathbb{E}[\Phi^{(\ell)}] - \mathbb{E}[\hat{\Phi}^\star]\|$ | 0.31 | 0.80 | 0.65 | 0.59 | 0.69 |
| $\|\sigma(\Phi^{(\ell)}) - \sigma(\hat{\Phi}^\star)\|$ | 20.00 | 2.25 | 1.93 | 5.55 | 7.30 |
| Test. Error | | | **7.60** | | |

