# OpenReview forum: "Simulated+Unsupervised Learning With Adaptive Data Generation and Bidirectional Mappings"
_ICLR.cc/2018/Conference — Accept (Poster)_

### Official Review · AnonReviewer1 · 2017-11-25
**This paper considers a generative approach in semi-supervised setting, which is basically a combination of cycleGAN and the Apple's S+U leaning.**

**Rating:** 6
**Confidence:** 3

**Review:**

* sec.2.2 is about label-preserving translation and many notations are introduced. However, it is not clear what label here refers to, and it does not shown in the notation so far at all. Only until the end of sec.2.2, the function F(.) is introduced and its revelation - Google Search as label function is discussed only at Fig.4 and sec.2.3.
* pp.5 first paragraph: when assuming D_X and D_Y being perfect, why L_GAN_forward = L_GAN_backward = 0? To trace back, in fact it is helpful to have at least a simple intro/def. to the functions D(.) and G(.) of Eq.(1).
* Somehow there is a feeling that the notations in sec.2.1 and sec.2.2 are not well aligned. It is helpful to start providing the math notations as early as sec.2.1, so labels, pseudo labels, the algorithm illustrated in Fig.2 etc. can be consistently integrated with the rest notations.
* F() is firstly shown in Fig.2 the beginning of pp.3, and is mentioned in the main text as late as of pp.5.
* Table 2: The CNN baseline gives an error rate of 7.80 while the proposed variants are 7.73 and 7.60 respectively. The difference of 0.07/0.20 are not so significant. Any explanation for that?
Minor issues:
* The uppercase X in the sentence before Eq.(2) should be calligraphic X

---

> ### Author Response · Authors · 2017-12-24
> **Responses**
>
> Dear Reviewer1,
>
> First of all, we really appreciate your constructive review and detailed comments on our work. We would like to share our responses to the concerns raised by the reviewer.
>
> 1. pp.5 first paragraph: when assuming D_X and D_Y being perfect, why L_GAN_forward = L_GAN_backward = 0? To trace back, in fact it is helpful to have at least a simple intro/def. to the functions D(.) and G(.) of Eq.(1).
> => Assume a perfect discriminator that always outputs 1 if the image is from the target (real) domain and 0 if it is from the source (fake) domain, regardless of how precise the generators are. Then, the standard GAN loss becomes 0 since L_{GAN}(X,Y) = E_{Y}[log(1)] + E_{X}[log(1-0)] = 0. To make the descriptions clearer, we added the definition of G() to Sec 2.1 along with other notations, and the definition of D() before equation (1).
>
> 2. Somehow there is a feeling that the notations in sec.2.1 and sec.2.2 are not well aligned. It is helpful to start providing the math notations as early as sec.2.1, so labels, pseudo labels, the algorithm illustrated in Fig.2 etc. can be consistently integrated with the rest notations.
> => We appreciate your great suggestion. As mentioned above, we included a separate subsection on notations and definitions. Further, we modified Sec. 2 to align the notations used in the two different sections.
>
> 3. F() is firstly shown in Fig.2 the beginning of pp.3, and is mentioned in the main text as late as of pp.5.
> => We fixed this.
>
> 4. Minor issues: * The uppercase X in the sentence before Eq.(2) should be calligraphic X
> => We fixed this.
>
> 5. Table 2: The CNN baseline gives an error rate of 7.80 while the proposed variants are 7.73 and 7.60 respectively. The difference of 0.07/0.20 are not so significant. Any explanation for that?
> => We would like to first clarify that the CNN baseline alone gives error rate of 11.2. The scheme that achieves the error rate of 7.80 is not a simple CNN baseline but the SimGAN (CVPR’17) approach (forward mapping + training a predictor on the forward-translated images), which is the previous state-of-the-art. With regards to the significance of the improvements, we believe that the error rate achieved by our algorithm is very close to the best error rate that one can hope for. Hence, in this regime, improving the state-of-the-art performance even by a small margin requires much efforts.
>
> Thanks,
> Authors

---

### Official Review · AnonReviewer3 · 2017-11-28
**Review: Some interesting ideas, some missing related work and comparisons**

**Rating:** 6
**Confidence:** 4

**Review:**

Review, ICLR 2018, Simulated+Unsupervised Learning With Adaptive Data Generation and Bidirectional Mappings

Summary:

The paper presents several extensions to the method presented in SimGAN (Shirvastava et al. 2017).
First, it adds a procedure to make the distribution of parameters of the simulation closer to the one in real world images. A predictor is trained on simulated images created with a manually initialized distribution. This predictor is used to estimate pseudo labels for the unlabeled real-world data. The distribution of the estimated pseudo labels is used produce a new set of simulated images. This process is iterated.
Second, it adds the idea of cycle consistency (e.g., from CycleGAN) in order to counter mode collapse and label shift.
Third, since cycle consistency does not fully solve the label shift problem, a feature consistency loss is added.
Finally, in contrast to ll related methods, the final system used for inference is not a predictor trained on a mix of real and “fake” images from the real-world target domain. Instead the predictor is trained purely on synthetic data and it is fed real world examples by using the back/real-to-sim-generator (trained in the conjunction with the forward mapping cycle) to map the real inputs to “fake” synthetic ones.

The paper is well written. The novelty is incremental in most parts, but the overall system can be seen as novel.

In particular, I am not aware of any published work that uses of the (backwards) real-to-sim generator plus sim-only trained predictor for inference (although I personally know several people who had the same idea and have been working on it). I like this part because it perfectly makes sense not to let the generator hallucinate real-world effects on rather clean simulated data, but the other way around, remove all kinds of variations to produce a clean image from which the prediction should be easier.

The paper should include Bousmalis et al., “Unsupervised Pixel-Level Domain Adaptation With Generative Adversarial Networks”, CVPR 2017 in its discussion, since it is very closely related to Shirvastava et al. 2017.

With respect to the feature consistency loss the paper should also discuss related work defining losses over feature activations for very similar reasons, such as in image stylization (e.g. L. A. Gatys et al. “Image Style Transfer Using Convolutional Neural Networks” CVPR 2016, L. A. Gatys et al. “Controlling Perceptual Factors in Neural Style Transfer” CVPR 2017), or the recently presented “Photographic Image Synthesis with Cascaded Refinement Networks”, ICCV 2017.
Bousmalis et al., “Using Simulation and Domain Adaptation to Improve Efficiency of Deep Robotic Grasping”, arXiv:1709.07857, even uses the same technique in the context of training GANs.

Adaptive Data Generation:
I do not fully see the point in matching the distribution of parameters of the real world samples with the simulated data. For the few, easily interpretable parameters in the given task it should be relatively easy to specify reasonable ranges. If the simulation in some edge cases produces samples that are beyond the range of what occurs in the real world, that is maybe not very efficient, but I would be surprised if it ultimately hurt the performance of the predictor.
I do see the advantage when training the GAN though, since a good discriminator would learn to pick out those samples as generated. Again, though, I am not very sure whether that would hurt the performance of the overall system in practice.

Limiting the parameters to those values of the real world data also seems rather restricting. If the real world data does not cover certain ranges, not because those values are infeasible or infrequent, but just because it so happens that this range was not covered in the data acquisition, the simulation could be used to fill in those ranges.
Additionally, the whole procedure of training on sim data and then pseudo-labeling the real data with it is based on the assumption that a predictor trained on simulated data only already works quite well on real data. It might be possible in the case of the task at hand, but for more challenging domain adaptation problem it might not be feasible.
There is also no guarantee for the convergence of the cycle, which is also evident from the experiments (Table 1. After three iterations the angle error increases again. (The use of the Hellinger distance is unclear to me since it, as explained in the text, does not correspond with what is being optimized). In the experiments the cycle was stopped after two iterations. However, how would you know when to stop if you didn’t have ground truth labels for the real world data?

Comparisons:
The experiments should include a comparison to using the forward generator trained in this framework to train a predictor on “fake” real data and test it on real data (ie. a line “ours | RS | R | ?” in Table 2, and a more direct comparison to Shrivastava). This would be necessary to prove the benefit of using the back-generator + sim trained predictor.


Detailed comments:
* Figure 1 seems not to be referenced in the text.
* I don’t understand the choice for reduction of the sim parameters. Why was, e.g., the yaw and pitch parameters of the eyeball set equal to those of the camera? Also, I guess there is a typo in the last equality (pitch and yaw of the camera?).
* The Bibliography needs to be checked. Names of journals and conferences are inconsistent, long and short forms mixed, year several times, “Proceedings” multiple times, ...

---

> ### Author Response · Authors · 2017-12-24
> **Responses**
>
> Dear Reviewer3,
>
> First of all, we really appreciate your constructive review and detailed comments on our work. We would like to share our responses to the concerns raised by the reviewer.
>
> 1. In particular, I am not aware of any published work that uses of the (backwards) real-to-sim generator plus sim-only trained predictor for inference…
> => We thank the reviewer for constructive feedback. We agree that our backward approach has some advantages over the forward approach exactly due to the reasons the reviewer mentioned.
>
> 2-1. The paper should include Bousmalis et al…
> 2-2. With respect to the feature consistency loss the paper should also discuss related work …
> => We thank the reviewer for sharing with us the key reference on the S+U learning and the works that proposed the feature consistency. We added them in our revision.
>
> 3. Adaptive Data Generation: I do not fully see the point in matching the distribution of parameters of the real world samples with the simulated data..
> => We agree that having samples that are beyond the range of real sample should not hurt the prediction performance. As the reviewer pointed out, the adaptive data generation part is for generating synthetic samples in a “sample-efficient” way. To see this, consider the following genie-aided experiments. The first label distribution of the simulator is set such that the mean and variance match those of the true label distribution, measured from the test set. The second one is set such that the mean is the same but the variance is twice larger than that of the ture one. (Note that this also generates all possible labels.) We ran our algorithms with these label distributions, and observed that the first one achieves the error rate of “7.74” and the second one achieves “8.88”. As the reviewer expected, we believe that the gap will diminish as the dataset size grows.
>
> 4. Limiting the parameters to those values of the real world data also seems rather restricting…
> => If the real data provided in the training set is not representative enough, our approach may not be able to generalize well on unseen data. One may address this limitation by incorporating prior knowledge about the label distributions or by manually tuning the parameters. A thorough study is needed to understand how one could obtain diverse synthetic images via such methods in a systematic way. We added a remark on this in the revised draft.
>
> 5. Additionally … the assumption that a predictor trained on simulated data only already works quite well on real data …
> => Indeed, our adaptive data generation method assumes that the predictor trained on simulated data works quite well on real data. Hence, if the predictor trained solely on simulated data provide completely wrong pseudo-labels, matching the synthetic label distribution with the pseudo-label distribution may not be helpful at all. For instance, when we pseudo-labeled the images in the SVHN dataset using a digit classifier that is trained on the MNIST dataset, our first stage failed to refine the synthetic label distribution. It is an interesting open question whether or not one can devise a similar adaptive data generation method for such cases. We added a remark on this limitation in the revised draft.
>
> 6. There is also no guarantee for the convergence of the cycle …  However, how would you know when to stop …?
> => Yes, the convergence is not guaranteed, and hence one needs to choose the right result to proceed with. Actually, we mentioned in the first manuscript that we made use of a small validation set (1% of the test data set) consisting of real data with labels to do so. For clarity, we also added the validation errors to the table.
>
> 7. The use of the Hellinger distance is unclear to me since it, as explained in the text, does not correspond with what is being optimized.
> => We used the Hellinger distance for quantifying the progress of the adaptive data generation algorithm. To clarify this, we added the sequence of means and variances to Table 1 instead of the Hellinger distance.
>
> 8. The experiments should include a comparison to using the forward generator trained in this framework to train a predictor on “fake” real data and test it on real data (ie. a line “ours | RS | R | ?” in Table 2, and a more direct comparison to Shrivastava)….
> => We added the performance corresponding to “ours | RS | R” to Table 2: It achieves a performance that is slightly better than the SimGAN’s one but worse than that of our back-generator.
>
> 9. I don’t understand the choice for reduction of the sim parameters…
> => Our internal experiments (not reported in the earlier draft) revealed that the choice of parameter reduction barely affects the performance of the algorithm. For clarification, in the revision (see appendix), we added experimental results comparing two different reduction methods.
>
> 10. Figure 1 not referenced  & A typo in the last equality & The Bibliography needs to be checked
> => We fixed them.
>
> Thanks,
> Authors

---

### Official Review · AnonReviewer2 · 2017-11-29
**Interesting application of GAN type training. But not good enough.**

**Rating:** 3
**Confidence:** 4

**Review:**

General comment:

This paper proposes a GAN-based method which learns bidirectional mappings between the real-data and the simulated data. The proposed methods builds upon the CycleGAN and the Simulated+Unsupervised (S+U) learning frameworks. The authors show that the proposed method is able to fully leverage the flexibility of simulators by presenting an improved performance on the gaze estimation task.

Detailed comments:

1. The proposed method seems to be a direct combination of the CycleGAN and the S+U learning. Firstly, the CycleGAN propose a to learn a bidirectional GAN model between for image translation. Here the author apply it by "translating" the the simulated data to real-data. Moreover, the mapping from simulated data to the real-data is learned, the S+U learning framework propose to train a model on the simulated data.

Hence, this paper seems to directly apply S+U learning to CycleGAN. The properties of the proposed method comes immediately from CycleGAN and S+U learning. Without deeper insights of the proposed method, the novelty of this paper is not sufficient.

2. When discussing CycleGAN, the authors claim that CycleGAN is not good at preserving the labels. However, it is not clear what the meaning of preserving labels is. It would be nice if the authors clearly define this notion and rigorously discuss why CycleGAN is insufficient to reach such a goal and why combining with S+U learning would help.

3. This work seems closely related to S+U learning. It would be nice if the authors also summarize S+U learning in Section 2, in the similar way they summarize CycleGAN in Section 2.2.

4. In Section 2.2, the authors claim that the Cycle-consistency loss in CycleGAN is not sufficient for label preservation. To improve, they propose to use the feature consistency loss. However, the final loss function also contains this cycle-consistency loss. Moreover, in the experiments, the authors indeed use the cycle-consistency loss by setting \lambda_{cyc} = 10. But the feature consistency loss may not be used by setting \lambda_{feature} = 0 or 0.5. From table Two, it appears that whether using the feature-consistency loss does not have significant effect on the performance.

It would be nice to conduct more experiments to show the effect of adding the feature-consistent loss. Say, setting \lambda_{cyc} = 0 and try different values of \lambda_{feature}. Otherwise it is unclear whether the feature-consistent loss is necessary.

---

> ### Author Response · Authors · 2017-12-02
> **Responses**
>
> Dear Reviewer2,
>
> First of all, we really appreciate your constructive review and detailed comments on our work. We would like to share our responses to the concerns raised by the reviewer.
>
> 1. The proposed method seems to be a direct combination of the CycleGAN and the S+U learning…. Without deeper insights of the proposed method, the novelty of this paper is not sufficient.
>
> => While we fully agree that some components of our framework are largely inspired by the CycleGAN and the S+U learning, our framework has its own novelty as follows.
> 1) It starts with a novel adaptive data generation process, which we observed necessary to achieve the state-of-the-art performances in our own experiments. To see this, we added the test errors to Table 1 in the revision. One can observe that one cannot achieve the state-of-the-art performances without having a “good” synthetic label distribution, which can be obtained by our adaptive data generation process.
> 2) Our approach is different from the traditional S+U learning framework: In the original S+U learning framework, the synthetic data is mapped to the real data, and then a predictor is trained on the translated data. In our work, we do not train our predictors after we learn the bidirectional mapping: Instead, we simply map test images to the synthetic domain, and directly apply predictors, which are trained solely with the synthetic data set.
> Our approach has a two-fold advantages over the traditional framework. First, we observe that our backward approach can achieve an improved prediction performance. To see this, we also added the test error measure with the traditional forward mapping approach, which is worse than our backward mapping approach. Further, our approach also has a significant saving in terms of computation over the traditional S+U learning since one does not have to retrain predictors for each target domain. (Having one good predictor trained on the synthetic domain suffices!)
>
> 2. When discussing CycleGAN, the authors claim that CycleGAN is not good at preserving the labels. However, it is not clear what the meaning of preserving labels is. It would be nice if the authors clearly define this notion and rigorously discuss why CycleGAN is insufficient to reach such a goal and why combining with S+U learning would help.
> => According to the reviewer’s comment, we made the following changes in how we describe the label-loss problem and our approach to mitigate the problem. We added a subsection (Sec 2.1) where we define notations and introduce new terms. We also added references to several prior works, which proposed a similar concept called ‘content representation’ or ‘feature matching’ for other related tasks. Further, we would like to clarify that we are proposing the use of “feature consistency loss” to address this challenge.
>
> 3. This work seems closely related to S+U learning. It would be nice if the authors also summarize S+U learning in Section 2, in the similar way they summarize CycleGAN in Section 2.2.
> => We now added more detailed description of the original S+U learning papers to Sec 2.4 in the revision.
>
> 4. In Section 2.2, the authors claim that the Cycle-consistency loss in CycleGAN is not sufficient for label preservation. To improve, they propose to use the feature consistency loss. However, the final loss function also contains this cycle-consistency loss.
> => First of all, we would like to clarify the following. In this work, our claim was that the cycle-consistency loss “alone” may not preserve labels well, and hence we propose to use it together with “feature-consistency loss”. That is, we are proposing to use both of them. For clarification, we revised the relevant descriptions.
>
> 5. Moreover, in the experiments, the authors indeed use the cycle-consistency loss by setting \lambda_{cyc} = 10. But the feature consistency loss may not be used by setting \lambda_{feature} = 0 or 0.5. From table Two, it appears that whether using the feature-consistency loss does not have significant effect on the performance. It would be nice to conduct more experiments to show the effect of adding the feature-consistent loss. Say, setting \lambda_{cyc} = 0 and try different values of \lambda_{feature}. Otherwise it is unclear whether the feature-consistent loss is necessary.
> => According to the reviewer’s comment, we included additional experimental results to see the roles of \lambda_{feature} and \lambda_{cycle}. More specifically, we run our algorithm with different combinations of \lambda_{feature} \in {0, 0.1, 0.5, 1.0} and \lambda_{cycle} \in {0,1,5,10,50}. As a result, we observe that setting “\lambda_{feature} = 0.5, \lambda_{cycle} = 10” achieved the best performance among the tested cases, proving the necessity of using both consistency terms. For details, see the experimental results in the appendix of the revision.
>
> Thanks,
> Authors

---

### Author Response · Authors · 2017-12-24
**Our common response for all reviewers**

Dear reviewers,

First of all, we really appreciate your constructive review and detailed comments on our work. Please find our detailed responses below. We also uploaded our revised manuscript where the reviewers' comments and feedback are reflected. Some of the major changes are colored in blue to help the reviewer locate them.

Please let us know if the reviewers have any further questions or comments.

Thanks,
Authors

---

### Decision · Program_Chairs · 2018-01-29
**ICLR 2018 Conference Acceptance Decision**

**Decision:**

Accept (Poster)

**Comment:**

Split opinions on paper: 6 (R1), 3 (R2), 6 (R3). Much of the debate centered on the novelty of the algorithm. R2 felt that the paper was a straight-forward combination of CycleGAN with S+U, while R3 felt it made a significant contribution. The AC has looked at the paper and the reviews and discussion. The topic is very interesting and topical. The experiments are ok, but would be helped a lot by including the real/synth car data currently in appendix B: seeing the method work on natural images is much more compelling. The approach still seems a bit incremental: yes, it's not a straight combination but the extra stuff isn't so profound. The AC is inclined to accept, just because this is an interesting problem.